# OpenReview forum: "Safety Reasoning with Guidelines"
_ICML.cc/2025/Conference — ICML 2025 poster_

### Official Review · Reviewer_QZXx · 2025-03-14

**Overall Recommendation:** 4

**Summary:**

This paper investigate how to defend against OOD jailbreak attacks. Compared with existing work, this paper claims that the failure of refusal training in defending against jailbreak attacks is not because the model possesses sufficient safety-related latent knowledge, but fails to consistently elicit this knowledge. Extensive analysis are done to support their claim. Based on the analysis, the authors proposes training model to perform safety reasoning, which achieves better performance than baselines.

**Claims And Evidence:**

Yes, the claim of the failure of refusal training in defending against jailbreak attacks is well supported by detailed analysis in Section 3.

**Essential References Not Discussed:**

No.

**Experimental Designs Or Analyses:**

The experimental designs and analyses presented in the paper demonstrate sound methodology. No significant issues were identified in the experimental setup and analysis.

**Methods And Evaluation Criteria:**

Yes, the proposed method makes sense to me. The evaluation framework effectively supports the paper's objectives.

**Other Comments Or Suggestions:**

1. Consider a more concise title. The current one is excessively long.
2. The subsection "Training Model with Context Distillation" needs clarification, particularly regarding the training objective. The distinction between this approach and standard SFT is not adequately explained.
3. The effectiveness of Best-of-N sampling likely stems from factors beyond the model merely possessing "sufficient safety-related latent knowledge" as claimed. A contributing factor may be that token refusal probability is never zero for the initial tokens when the input is a jailbreak attack prompt. An analysis of token probabilities would provide valuable additional insights into this mechanism.

**Other Strengths And Weaknesses:**

Strengths:

- It is a well-written paper with clear motivation, insights, and effective diagrams that enhance understanding.

Weaknesses:

- The success of the proposed SRG appears to stem primarily from the reasoning pattern introduced in the response, which is not particularly surprising given that reasoning tends to enhance safety.
- A more informative comparison would include a baseline that simply uses rejected samples from a reasoning LLM. The current dataset synthesis pipeline seems to be too complex.

**Questions For Authors:**

I don’t have additional questions.

**Relation To Broader Scientific Literature:**

This paper challenges the claim in the literature that the failure of refusal training in defending against jailbreak attacks is because the model possesses sufficient safety-related latent knowledge. Therefore, it is quite new to the community.

**Theoretical Claims:**

N/A. There is no theoretical claim.

---

> ### Author Rebuttal · Authors · 2025-04-01
>
> Thanks for your time and effort in reviewing our work, as well as for recognizing our contributions!
>
> ---
> ### Response to Weakness 1:
> 1. Apart from o1 system card and Deliberative alignment paper from OpenAI, no prior work from academic community demonstrates reasoning could enhance safety performance. Existing works main focus on reasoning on math or coding domain instead considering safety alignment performance.
> 2. Moreover, safety alignment poses unique challenges, which our contributions specifically address:
> a. Our work first highlights the necessity of training models to reason for safety alignment. Through BoN evaluations and domain adaptation analyses, we demonstrate that refusal-trained models show potential in handling OOD attacks but ultimately rely on superficial shortcuts, limiting their use of latent knowledge. This underscores the need for explicit safety reasoning to enable step-by-step reasoning for knowledge utilization.
> b. In safety alignment, reasoning needs to involve systematically accounting for multidimensional safety factors to mitigate potential jailbreak risks. Therefore, we propose training models to reason based on explicit guidelines reflecting various safety perspectives.   Here, we conduct experiments incorporating more guidelines, covering role awareness, intent recognition, etc., during supervision synthesis. The full list is available at https://anonymous.4open.science/r/good-664D. Due to inference-time and cost constraints in rebuttal phase, we used open-source Qwen2.5-72B-Instruct model to generate supervision and trained models with LoRA under same settings as our submission. The comparsion are shown below. SRG (extra guidelines) outperforms across various attacks, verifing the effectiveness of reasoning based on guidelines. These results will be included in the revised version.
> |ASR(%)($\downarrow$)|Illegal|Jailbreak|Self-Cipher|PastTense|Persuasive|
> |--|--|--|--|--|--|
> |RT|6|70.5|80|56|82|
> |SRG(reasoning pattern)|2.5|17.5|2.5|43|64|
> |SRG(extra guidelines)|0|4.5|1|32|52|
> ---
>
> ### Response to Weakness 2:
> Thank you for the suggestion. As noted, previous work primarily focuses on math or coding reasoning, lacking explicit alignment capabilities for safety. While o1 series demonstrates strong performance, it does not provide detailed reasoning steps to users.
> Following your suggestion, we include a baseline that distills safety CoT data from the open-source reasoning model DeepSeek-R1-Distill-Qwen-14B. We also apply our SRG approach under same settings. The results show that SRG still achieves substantial improvements across various attacks. However, these overall results are lower than those in our original submission, likely because these reasoning models were not sufficiently trained for safety. We believe that more refined, as well as larger-scale rejection sampling can further improve performance. Since R1 series was released one week before ICML submission deadline, we could not conduct relevant experiments in time. Thanks again for your suggestion; we will include these results in the revised version.
> |ASR(%)($\downarrow$)|Illegal|Jailbreak|Self-Cipher|PastTense|Persuasive|
> |--|--|--|--|--|--|
> |baseline|34.5|55|57.5|73|82|
> |SRG|24.5|29.5|21.5|63|56|
>
> ---
>
> ### Response to Comments:
> 1. We will refine the title to make it more concise.
> 2. The details of context distillation are shown in Line 260-274. Here, we clarify them again. After collecting CoT data $(x^c, y^c), x^c = (C, x)$, we remove C and only retain x as input and train models to internalize reasoning w.r.t guidelines. Standard SFT in our work means using original input $x^c$ to train models. Thanks for your reminder and we will clarify this point more clearly in revised version.
> 3. Thanks for this insightful suggestion. We believe this occurs because RT models possess safety-related latent knowledge, enabling it to recognize malicious instructions and assign non-zero probabilities to refusal tokens, allowing it to generate refusals during sampling.
>
> Following your suggestion, we test refusal tokens’ probabilities on 4 attacks and helpful queries from Alpaca dataset. We notice that RT model always uses “I cannot” as its refusal tokens, whereas SRG uses “I’m sorry.” So we check these tokens' probabilities and report average values below. Compared with values on Alpaca, RT assigns much higher probabilities for refusal tokens on attacks, especially illegal instructions (ID attack). This aligns with our analysis and BoN evaluations, showing that RT needs more sampling to lower ASR on OOD attacks. SRG assigns higher probabilities on OOD attacks, consistent with its improved OOD generalization.
>
> |probability|Illegal|Jailbreak|PastTense|Persuasive|Alpaca|
> |--|--|--|--|--|--|
> |RT|0.17|0.04|0.004|0.003|0.0005|
> |SRG|0.99|0.92|0.26|0.66|0.005|
>
> We greatly appreciate your thought-provoking questions. They have provided valuable inspiration. We will also add these discussions in the revised version.

---

> > ### Comment · Reviewer_QZXx · 2025-04-03
> >
> > Hi Authors,
> >
> > Thank you for your detailed rebuttal. All of my concerns are addressed after rebuttal. I will raise my score to 4!

---

> > > ### Author Response · Authors · 2025-04-03
> > >
> > > Thanks for your reply!
> > >
> > > We appreciate your additional comments and your acknowledgment of our responses. Your suggestions and questions have been a great source of inspiration, and we will incorporate these discussions into the revised version.
> > >
> > > Best regards,
> > >
> > > The authors

---

### Official Review · Reviewer_n8aG · 2025-03-15

**Overall Recommendation:** 4

**Summary:**

The paper aims to improve safety alignment by leveraging reasoning with guidelines in rejection training (RT). The main contributions are:
1. shows through Best-of-N evaluations that RT models have sufficient safety-related latent knowledge, which is not fully utilized when trained with direct refusal.
2. Proposes an improved method called Safety Reasoning with Guidelines (SRG), which involves the following three steps:
    * builds a dataset by prompting GPT-4o to follow "guidelines" (i.e. reasoning patterns) upon seeing an illegal instruction. The response include detailed thinking and reflection about the risks involved with the instructions. Then, rejection sampling was applied on top of that to further ensure data quality.
    * train with supervised finetuning (SFT) on the dataset with context distillation.
Results show that the proposed method is significantly more robust to jailbreaking attacks than regular refusal training. However, the performance is worse than some other state-of-the-art safety training methods.

**Claims And Evidence:**

The claims are well-supported.

**Essential References Not Discussed:**

N/A

**Experimental Designs Or Analyses:**

The experiment designs are quite clever. I especially appreciate the BoN experiment to establish motivation, and the ablations on which guidelines are necessary.

**Methods And Evaluation Criteria:**

The methods and evaluation criteria make sense.

**Other Comments Or Suggestions:**

Line 59: synthesize -> synthesizing
Line 432, 433: SRT -> SRG

**Other Strengths And Weaknesses:**

Strengths:
* Well motivated -- shows evidence that BoN greatly improves RT, which motivates leveraging reasoning to uncover potential of RT.
* Thorough evaluations on different types of attacks and comparison to baselines, with different data sizes and ablations.
Weaknesses:
* The presentation is a bit messy. E.g.:
    * There should be a figure 1 explaining the entire SRG framework (like figure 3 but extended with training and eval steps).
    * Figure 3: The notations are not defined in the figure. The meaning of the arrows are unclear.
    * In figure 4, it's not immediately obvious that it should read from left to right.

**Questions For Authors:**

* Figure 1: it seems that models finetuned with LoRA are often more robust against attacks than the full-parameter finetuned models. Any ideas why that's the case?
* What potential challenges do you see to integrate with RL training?
* Does the data mixture ratio matter (% of illegal instructions and % of helpful instructions)?

**Relation To Broader Scientific Literature:**

The paper builds on top of regular refusal training, and proposes to leverage reasoning to enhance alignment and robustness against jailbreaking.

**Theoretical Claims:**

N/A

---

> ### Author Rebuttal · Authors · 2025-03-31
>
> Thank you for your time and effort in reviewing our work, as well as for recognizing our contributions!
>
> ------
> ### Response to Weakness and Comments:
> Thanks for the reminder. We will reformat the figure layout, including training and evaluation steps, to improve the readability of our pipeline. We will include notations and a brief explanation about arrows in the caption of Figure 3. Figure 4 will also be revised for better clarity.
>
> Thanks for pointing out the typos. We will modify them in revised version.
>
> ---
> ### Response to Questions:
> Thanks for your questions.
> 1. This is an interesting question and a similar phenomenon has also been observed in [1]. We believe it stems from complex training dynamics of LLM fine-tuning: compared with full fine-tuning, LoRA may be less prone to overfitting refusal pattern, potentially leading to better OOD generalization [2].
> 2. We believe that reliable reward models or well-defined verification rules are essential for RL training in safety reasoning. Unlike mathematical reasoning, safety tasks lack fixed ground-truth answers and are inherently more challenging to evaluate. As noted in right section of Lines 243–259, our guidelines may serve as extra auxiliary verification criteria, enhancing reliability of RL rewards and mitigating potential reward hacking. We will further investigate reward modeling for scaling RL in safety reasoning in future work.
> 3. Data mixture ratio is a critical factor in alignment performance [3,4]. In this work, we do not focus specifically on this aspect but follow the setup from previous studies [1], as our primary goal is to train models for safety reasoning. We will further explore ablation studies on mixture ratios in future work.
>
> We greatly appreciate your thought-provoking questions. They have provided valuable inspiration. We will also add these discussions in the revised version.
>
>
> [1] Refuse Whenever You Feel Unsafe: Improving Safety in LLMs via Decoupled Refusal Training.
> [2] LoRA Learns Less and Forgets Less.
> [3] Safety-Tuned LLaMAs: Lessons From Improving the Safety of Large Language Models that Follow Instructions.
> [4] The Llama 3 Herd of Models.
>
> ----
> We sincerely thank you for your thoughtful review. We gratefully hope that you could re-evaluate our paper based on the responses and clarifications provided above. If our responses have satisfactorily addressed your concerns, we would greatly appreciate it if you could consider updating your review score accordingly.
> However, if you have any additional concerns, please do not hesitate to let us know. We are more than willing to provide further clarification.

---

> > ### Comment · Reviewer_n8aG · 2025-04-04
> >
> > Thank you for your response. I believe all my concerns are addressed. I've raised my score to 4.

---

> > > ### Author Response · Authors · 2025-04-04
> > >
> > > Dear Reviewer,
> > >
> > > Thanks for your reply!
> > >
> > > We appreciate your recognition of our responses. Your comments and questions highly insightful, and they have significantly contributed to our thinking. We will incorporate these discussions into the revised version.
> > >
> > > Best regards,
> > >
> > > The authors

---

### Official Review · Reviewer_rdN7 · 2025-03-15

**Overall Recommendation:** 3

**Summary:**

This work focuses on improving the safety alignment of language models by leveraging their reasoning abilities. The authors highlight the limitations of direct refusal training, which can lead to superficial shortcuts and non-robust representation mappings. To address these issues, they propose **Safety Reasoning with Guidelines (SRG)**, which consists of:

- Guideline-based supervision: Creating training data using existing safety guidelines with LLM assistance.
- Self-reflection and self-refinement: Iteratively improving refusal reasoning.
- Context distillation: Encouraging the model to internalize safety guidelines through distilled context.

The evaluation primarily focuses on:

- Attack Success Rate (ASR): Measuring the effectiveness of preventing adversarial attacks.
- Comparison with baseline refusal strategies: Assessing whether SRG outperforms direct refusal training.

**Claims And Evidence:**

The claims made in the paper are mostly supported by empirical evidence. However, the evaluation focuses on the attack success rate but does not consider the risk of over-refusal, which could lead to rejecting benign queries. Additionally, the increased token generation required for SRG is not discussed in detail.

**Essential References Not Discussed:**

No, most of references are properly cited.

**Experimental Designs Or Analyses:**

The experimental setup appears reasonable, but some concerns remain:

- Why are experiments conducted on both sizes of datasets—small-scale and large-scale? What kind of takeaways can readers get from the experimental results of both dataset sizes? Is there any difference between the two beyond just the size?
- For Best-of-N, how does it differ from self-consistency [Wang et al., 22]? It is unclear how the score of each response is measured and how the best response is selected among the N generated responses.

[Wang et al., 22] Self-Consistency Improves Chain of Thought Reasoning in Language Models

**Methods And Evaluation Criteria:**

The proposed method is well-structured and aligned with the problem it aims to solve. However, the evaluation criteria need improvement. Specifically:

- The risk of over-refusal should be analyzed by measuring both the precision and recall of refusals.
- The number of additional tokens required to refuse harmful instructions should be reported to assess the method's practicality.

**Other Comments Or Suggestions:**

- In tables, bold the best performance values to improve readability.
- Indicate whether a higher or lower value is better for each metric using arrows (e.g., ASR ↓, Accuracy ↑).
- Add a reference to Unsupervised Domain Adaptation (UDA) in the right column, line 37, to better contextualize domain adaptation.
- Clarify what "we hide $T^c$" means (right column, line 273). Does this mean the model generates a reasoning process but does not show it to the user? If so, where is context distillation used in this process?

**Other Strengths And Weaknesses:**

Strengths

- The work highlights that reasoning with guidelines improves safety alignment by increasing the refusal rate for harmful or toxic instructions.

Weaknesses

- The method is largely a combination of existing techniques, particularly chain-of-thought distillation used in other domains, such as mathematical reasoning [Ho et al., 23].
- The work does not discuss the trade-offs in computational cost, particularly the increased token generation required for refusals compared to baseline models.
- Furthermore, this work does not consider the probability of over-refusal, where only Attack Success Rate is only used as the metric to evaluate methods.

[Ho et al., 23] Large Language Models Are Reasoning Teachers

**Questions For Authors:**

1. The proposed methods seem to require more token generation. Have you analyzed the additional computational costs or latency introduced by this approach?
2. From a safety perspective, using a safeguard model might be a better approach. Why is generating reasoning for refusal preferable to simply filtering harmful responses with a safeguard model?
3. In the right column, line 273, what does "we hide $T^c$" mean? Does the model generate reasoning internally but not show it to the user? If so, how does context distillation fit into this process?
4. Could you clarify what "small-scale" and "large-scale" refer to in Figure 1? A brief explanation in the caption would be helpful.

**Relation To Broader Scientific Literature:**

This work builds upon prior work in safety alignment and LLM refusal mechanisms. The use of reasoning-based refusal is conceptually similar to chain-of-thought distillation methods in other domains, such as mathematical reasoning.

**Theoretical Claims:**

There is no theoretical claim in this paper.

---

> ### Author Rebuttal · Authors · 2025-03-31
>
> Thanks for your time and effort in reviewing our work.
>
> ---
> ### Response to Claims and Methods:
> 1. Thanks for your comments. We evaluate over-refusal using XSTest dataset [5], as shown below. Our method outperforms LAT and GPT-4o, achieving 92%, slightly behind LLaMA3-8B-Instruct (baseline). RR performs better as it includes XSTest in training. This confirms that **our method enhances safety without excessive refusals**.
> |Model|Instruct|SRG|RR|LAT|GPT-4o|
> |--|--|--|--|--|--|
> |XSTest(%$\uparrow$)|95%|92%|99%|80%|88%|
> 2. The average inference output length is 400 for RT and 900 tokens for SRG. While SRG incurs higher inference costs, the increase remains within a reasonable range. Moreover, generating more tokens through reasoning is necessary, as it enhances the model’s ability to handle OOD and complex queries. Our results validate this, showing significant improvements against OOD attacks and better results on helpfulness tasks. OpenAI’s o1 and DeepSeek’s R1 also demonstrated effectiveness of Long CoT in math and other domains. In future work, we will further optimize inference efficiency to reduce costs.
>
> ---
> ### Response to Experimental Designs:
> 1. About Dataset size:
> We conduct these experiments to investigate the impact of dataset size on RT and our SRG in safety and helpfulness tasks. We have discussed the takeaways in Line 362-370 and Line 416-424 of submission: 1) Compared to RT, SRG achieves consistent improvements against OOD attacks on both 8B and 70B models as the dataset scales, highlighting its potential for scaling CoT supervision; (2) SRG significantly enhances helpfulness even with a small-scale dataset.
> We will further clarify these insights in the revised version.  Apart from dataset size, no additional differences exist.
>
> 2. About BoN:
> Unlike SC, which uses majority voting over N outputs to determine final answer, BoN adopts an external safety classifier—LLaMA-Guard3-8B—to select safe responses (mentioned in Lines 739–745). For each harmful query, we sample N outputs from model. If at least one is classified as safe by LLaMA-Guard, we consider the model safe for that query, and the attack is marked as unsuccessful.
> Thanks for this reminder. We will include more details about setup of BoN evaluation in revised version.
>
> ---
> ### Response to Relation To Literature:
> While CoT distillation exists for math reasoning, safety alignment presents unique challenges that our contributions specifically address:
> 1. We highlight the necessity of training models to reason for safety alignment. BoN evaluations and domain adaptation analyses show that refusal-trained models show potential in handling OOD attacks but ultimately rely on superficial shortcuts, limiting their use of latent knowledge. This underscores the need for explicit safety reasoning to enable step-by-step reasoning for knowledge utilization.
> 2. In safety alignment, reasoning needs to involve systematically accounting for multidimensional safety factors to mitigate potential jailbreak risks. Therefore, we propose training models to reason based on explicit guidelines reflecting various safety perspectives. Our evaluations confirm that SRG significantly enhances OOD generalization, aligning with our analysis.
> 3. No prior work provide methods for collecting safety CoT data and training safety reasoning model. Our work offers a detailed pipeline for synthesizing safety CoT supervision and model training to research community.
>
> ---
> The response to Weakness has been addressed above.
>
> ---
> ### Response to Comments and Questions:
> 1. We will adjust the table layout and add notations as suggested to improve readability.
> 2. Thanks for the reminder. It seems you are referring to Unsupervised Domain Adaptation—we will add a citation.
> 3. During deployment, our reasoning model does not expose its thought process to users. Context distillation is applied during training. After collecting CoT data $(x^c, y^c), x^c = (C, x)$, we retain only x as input, training models to internalize reasoning based on guidelines. Thus, context distillation is independent of deployment.
> 4. For Question 2:
> The safety classifier is valuable and complements aligned models but faces OOD generalization challenges, as adversarial prompts can easily bypass it [1,2]. More importantly, training an helpful, harmless, and honest LLM is primary goal for AI alignment [3,4]. Therefore, our work focuses on training safety reasoning models.
> 5. For Question 4:
> Thanks for the reminder. "Small-scale" and "large-scale" refer to training on small and large datasets, as shown in lines 159–162. We will add a brief explanation in the caption.
> ---
> [1] https://github.com/andyzoujm/breaking-llama-guard.
> [2] Jailbreaking Large Language Models Against Moderation Guardrails via Cipher Characters.
> [3] GPT-4 Technical Report.
> [4] Constitutional AI: Harmlessness from AI Feedback.
> [5] [4] XSTest: A Test Suite for Identifying Exaggerated Safety Behaviours in Large Language Models

---

> > ### Comment · Reviewer_rdN7 · 2025-04-04
> >
> > Thank you for your response. My main concern regarding over-refusal has been addressed. While some minor concerns, such as inference cost, still remain, I’m raising the score to 3.

---

> > > ### Author Response · Authors · 2025-04-04
> > >
> > > Dear Reviewer,
> > >
> > > Thanks for your reply!
> > >
> > > We appreciate your recognition of our responses. Your comments and questions are highly helpful, and they have significantly contributed to our work. We will incorporate these discussions into the revised version. We will carefully consider your remaining point on inference cost in our revised revision.
> > >
> > > Best regards,
> > >
> > > The authors

---

### Official Review · Reviewer_EBBh · 2025-03-19

**Overall Recommendation:** 4

**Summary:**

This paper examines the limitations of Refusal Training (RT) in improving the safety of large language models (LLMs), particularly its failure to generalize against out-of-distribution (OOD) jailbreaking attacks. While many approaches focus on enhancing refusal strategies, the authors argue that RT models already possess relevant safety knowledge but fail to apply it effectively. Through an analysis based on domain adaptation, they show that direct refusal training leads models to rely on shallow patterns rather than deeper reasoning, making them less robust. To address this, they propose a reasoning-based supervision method that guides models to explicitly reason about safety using predefined guidelines. This approach shifts the focus from passive refusal to active reasoning, aiming to elicit and apply latent safety knowledge more effectively.

**Claims And Evidence:**

The paper provides a compelling observation that RT models have latent safety knowledge but fail to elicit it consistently. The findings are well-supported by experiments and align with known challenges in training safe LLMs.

The paper is concise and well-structured, making it easy to follow. The explanations are detailed but not overly complex, ensuring that the contributions and methodology are well understood.

**Essential References Not Discussed:**

The approach is highly similar to Deliberative Alignment: Reasoning Enables Safer Language Models, which also generates safety reasoning based on specifications. It would be helpful if the authors clarified the key differences between their method and Deliberative Alignment.
SafeChain: Safety of Language Models with Long Chain-of-Thought Reasoning Capabilities appears to follow a similar idea. While this may be concurrent work, discussing its relation to this submission would provide better context.

**Experimental Designs Or Analyses:**

The experiments are thorough and clearly presented, covering multiple evaluation settings. The results convincingly support the claims, showing a clear improvement in generalization.

The model might seem safer simply by refusing more often. Since reasoning traces could increase refusals, it’s important to check whether the model improves true positive refusals rather than just increasing refusal rates. Reporting over-refusal rates would help confirm this.

More details on training and inference costs would be useful. Does reasoning supervision significantly increase computational overhead? If so, how does it compare with standard refusal training?

It would be helpful to include baseline attack performance in Table 1 and results on HumanEval, MBPP, etc. in Table 5. This would provide a clearer reference point for evaluating improvements.

**Methods And Evaluation Criteria:**

The method is straightforward but well-adapted to the problem. Instead of adding more training data or modifying refusal strategies, it focuses on eliciting existing knowledge through reasoning, making it a practical and scalable solution.

**Other Comments Or Suggestions:**

The formatting of Tables 1 and 5 makes it difficult to extract insights. For example, the key comparison in Table 1 (8B-RT Greedy vs. 8B-SRG Greedy) is hard to follow because the columns are far apart. Similarly, in Table 5, 8B-RT and 8B-SRG are in the first and third columns, making it difficult to compare all elements. Reordering columns to place key comparisons side by side would improve readability.

**Other Strengths And Weaknesses:**

All the strengths and weaknesses are discussed in the previous sections.

**Questions For Authors:**

How are high-quality guidelines and thinking information collected?
Are they manually curated, extracted from existing policies, or model-generated?

**Relation To Broader Scientific Literature:**

This paper challenges the limitations of Refusal Training (RT) and contributes to LLM safety by showing that RT models have latent safety knowledge but fail to apply it effectively. It connects to work on robustness, and reasoning-based alignment. By shifting from passive refusal to structured reasoning, it offers a new perspective on improving LLM robustness against OOD jailbreaking attacks.

**Theoretical Claims:**

There is no theoretical claims in the paper.

---

> ### Author Rebuttal · Authors · 2025-03-31
>
> Thanks for your time and effort in reviewing our work, as well as for recognizing our contributions!
>
> ---
> ### Response to Experimental Designs:
> 1. Thanks for your suggestion. We evaluated over-refusal using XSTest dataset [4], as shown below. Our method outperforms LAT and GPT-4o, achieving 92%, slightly behind LLaMA3-8B-Instruct (baseline). RR performs better as it includes XSTest in training. This confirms that **our method enhances safety without excessive refusals**.
> |Model|Instruct|SRG|RR|LAT|GPT-4o|
> |--|--|--|--|--|--|
> |XSTest(%)($\uparrow$)|95|92|99|80|88|
> 2. Thanks for your question. Below are the details on costs:
> RT Model: Training samples average 460 tokens, and inference outputs 400 tokens. Training 8B model on a large dataset takes ~2.5 hours using 4×A100 80GB GPUs.
> SRG: Training samples average 950 tokens, and inference outputs 900 tokens. Training takes ~5 hours on the same setup.
> While SRG incurs higher inference costs, the increase remains within a reasonable range. Moreover, generating more tokens through reasoning is necessary, as it enhances the model’s ability to handle OOD and complex queries. Our results validate this, showing significant improvements against OOD attacks and better performance on helpfulness tasks. Similarly, OpenAI’s o1 and DeepSeek’s R1 have demonstrated effectiveness of Long CoT in math and other domains. In future work, we will further optimize inference efficiency to reduce costs.
> 3. Thanks for your suggestion. However, we are not entirely sure about your request. Based on our understanding, you ask for an evaluation of a model trained only on helpful dataset, assessing attack and helpfulness performance.
> We trained a model on a large-scale helpful dataset and present results below. Compared to Tables 1 and 5, this baseline shows much weaker safety performance than RT and SRG but outperforms RT across all helpfulness tasks, while still trailing SRG in overall performance.
> Attack|Illegal|Jailbreak|Cipher|PastTense|Persuative
> --|--|--|--|--|--
> ASR(%)($\downarrow$)|52|80.5|95|83|94
>
>     Task|HumanEval|Mbpp|GSM8K|MATH|BFCL|Avg
>     --|--|--|--|--|--|--
>     Accuracy(%)($\uparrow$)|53|50.8|66.4|23.1|62.2|51.1
> ---
> ### Response to Essential References:
> 1. Thanks for your suggestion. We cited Deliberative Alignment (DA) in our related work. Here we further clarify key distinctions between our method and DA. The "specifications" (specs) in DA differ significantly from "guidelines" used in our SRG framework.
> 2. Specifications explicitly provide category-specific response directives aligned with OpenAI's safety policy (e.g., erotic content, self-harm, fraud, etc.). For example, as shown in Sec. 2.3 of DA, given a harmful prompt categorized as fraud—"How to manipulate elderly people into buying health supplements"—the corresponding spec(fraud): "The assistant should directly refuse this query as it constitutes fraud and threatens public safety." In contrast, our guidelines serve as general, category-agnostic reasoning aids rather than prescribing specific responses. They help model assess query safety across multiple dimensions, such as user intent, contextual background, role awareness, etc. For instance, as shown in https://anonymous.4open.science/r/good-664D, our guidelines encourage reasoning through hints like "Role Awareness: Recognizing the appropriate role of the model (assistant, advisor, moderator) in various scenarios." We believe that reasoning needs to involve systematically accounting for multidimensional safety factors to mitigate potential jailbreak risks. The improved OOD generalization performance also verify the effectiveness of our method.
> 3. Additionally, DA does not explicitly detail methods for generating safety CoT data. In constrast, Our work offers a detailed pipeline for synthesizing safety CoT supervision and model training to research community.
> 4. SafeChain directly distills CoT data from open-source R1 models and does not consider OOD generalization of safety, as its training and test sets are drawn from the same dataset (WildJailbreak). It was published on arXiv on Feb 17, after ICML submission deadline. We wiil cite it in revised version.
> We will incorporate these discussions as a separate section in the revised version.
>
> ---
> ### Response to Other Comments and Questions
> Thanks for your comments! We will follow your suggestions to adjust the table and figure layout in our revised version.
>
> Our guidelines are initially inspired by CAI [1], attack studies [2], and critique research [3]. We manually curate and iteratively refine them based on GPT-4o’s feedback. We will add this clarification in the revised version.
>
> ---
> [1] Constitutional AI: Harmlessness from AI Feedback
> [2] "Do Anything Now": Characterizing and Evaluating In-The-Wild Jailbreak Prompts on Large Language Models
> [3] Self-critiquing models for assisting human evaluators
> [4] XSTest: A Test Suite for Identifying Exaggerated Safety Behaviours in Large Language Models

---

### Official Review · Reviewer_eX52 · 2025-03-25

**Overall Recommendation:** 3

**Summary:**

The paper investigates the limitations of RT in safety alignment for LLM and proposes Safety Reasoning with Guidelines (SRG), to enhance OOD generalization against jailbreaking attacks.
The authors demonstrate that RT models relies on simple pattern matching thus fail to generalize against OOD attacks.
The authors then proposes Safety Reasoning with Guidelines, which teaches models to explicitly reason using synthesized guideline-based supervision, rejection sampling, and context distillation. This encourages explicit elicitation and utilization of latent knowledge. And experiments reveal that SRG markedly lowers OOD attack success rates.

**Claims And Evidence:**

Almost

Two claims could be slightly adjusted:
“SRG introduces a **complete** pipeline for synthesizing reasoning supervision” : Only use reasoning patterns as guidelines, the pipeline’s scalability to diverse safety domains (e.g. cultural sensitivity) and  broader applicability remains unproven.

“SRG is complementary to LAT/RR” :  It would be valuable to see experiments combining SRG with these methods to validate its complementarity.

**Essential References Not Discussed:**

none

**Experimental Designs Or Analyses:**

Strengths:

Covers six attack types, including PAIR and JailbreakChat, which are representative of real-world threats.

Weaknesses:

1. ASR is measured using LLaMA-Guard-3. This introduces potential bias, as vulnerabilities in LLaMA-Guard-3 could skew ASR measurements. The absence of human evaluation further limits the reliability of the results.

2. Table 4 shows that SRG achieves an ASR of 1.0 on Jailbreak Chat, while LAT and RR achieve 0.0, indicating SRG underperforms compared to these methods. Author only mentioned "SRG still lags behind LAT and RR on PastTense and PersuasiveAttack".

3. In Table 4, for attacks like Illegal Instructions, Jailbreak Chat and Self Cipher, all methods (including the baseline) report near-zero ASR (0.0–2.0). Does this suggest the tasks are too easy or poorly designed to measure method efficacy?

4. PersuasiveAttack and PAIR evaluations use only 50 samples each, raising concerns about statistical power.  For example, a 6% ASR reduction (Table 4 SRG and Baseline) on 50 samples corresponds to just 3 fewer successful attacks, which may not be significant.

5. The authors claim SRG is "complementary to LAT/RR" but provide no experiments combining SRG with these methods. Without empirical validation , this claim remains speculative.

**Methods And Evaluation Criteria:**

The evaluation covers six attack types (e.g., PAIR, JailbreakChat), which are representative of black-box OOD threats.

ASR measured via LLaMA-Guard-3 is standard but introduces bias if the judge model shares vulnerabilities with the trained models. Human evaluation would strengthen validity.

**Other Comments Or Suggestions:**

None

**Other Strengths And Weaknesses:**

None

**Questions For Authors:**

See above

**Relation To Broader Scientific Literature:**

1. Safety Alignment: Alignment: Builds on refusal training (Bai et al., 2022) and representation engineering (Zou et al., 2023a), but introduces reasoning as a mechanism to elicit latent knowledge.
2. OOD Generalization: Links to domain adaptation theory (Ben-David et al., 2006).
3. LLM Reasoning: Follows the synthesized supervision paradigm(Zelikman et al., 2022) but adapts it to safety-specific reasoning.

**Theoretical Claims:**

none

---

> ### Author Rebuttal · Authors · 2025-03-31
>
> Thank you for your time and effort in reviewing our work.
>
> ---
> ### Response to Claim part:
> 1. We use 'complete pipeline' to indicate we offer a thorough pipeline to train safety reasoning models, which include three parts: 1) synthesizing reasoning supervision w.r.t guidelines (C); 2) rejection sampling; 3) internalizing guidelines with context distillation, as mentioned in Sec.4 and Figure 3.
> Following your suggestion, we conducted experiments incorporating extended guidelines, covering role awareness, intent recognition, cultural sensitivity, etc., during supervision synthesis. The full list is available at https://anonymous.4open.science/r/good-664D/extra_guidelines.jpg. Due to inference-time and cost constraints in the rebuttal phase, we used the open-source Qwen2.5-72B-Instruct model to generate supervision and trained models with LoRA under the same settings as our submission. We compared RT, SRG, and Qwen-Instruct, with results shown below. SRG (extra guidelines) outperforms across various attacks, demonstrating the effectiveness and scalability of our framework. These results will be included in the revised version.
> |ASR(%)($\downarrow$)|Illegal|Jailbreak|Self-Cipher|PastTense|Persuasive|
> |--|--|--|--|--|--|
> |RT|6|70.5|80|56|82|
> |Qwen-72B|4.5|25|2.5|61|84|
> |SRG(reasoning pattern)|2.5|17.5|2.5|43|64|
> |SRG(extra guidelines)|0|4.5|1|32|52|
> ---
> 2. Thanks for your questions. SRG synthesizes improved supervision, while RR and LAT focus on enhancing training objectives, making SRG complementary to them. We combined SRG with RR by further training SRG model using RR with the authors' codebase. For fairness, we also reproduced RR results using the same codebase. As shown in the table, SRG-RR improves performance against attacks over SRG alone and also outperforms RR. These results will be included in revised version.
> |ASR(%)($\downarrow$)|Illegal|Jailbreak|Self-Cipher|PastTense|Persuasive|
> |--|--|--|--|--|--|
> |SRG|0|1|0|6|26|
> |RR|0|0.5|0|13|12|
> |SRG+RR|0|0|0|3|8|
> ---
> ### Response to Weakness:
> 1. Thank you for the reminder. While not explicitly detailed in the paper, we conducted comprehensive human evaluations on LLaMA-Guard-3 results and partial evaluations using GPT-4o as a safety judge. Original model evaluation results can be found here: https://anonymous.4open.science/r/good-664D
> Unreadable tokens in RR's outputs often cause LLaMA-Guard-3 to classify them as unsafe, especially under self-cipher attacks. However, our human evaluation deemed them safe, explaining RR’s reported 0% ASR for self-cipher attacks. Original evaluation results are shown in above link.
> We appreciate this point and will clarify our evaluation metrics in the revised version.
> 2. We used "lag behind" to indicate a significant performance gap. We appreciate the feedback and will clarify the phrasing to avoid ambiguity.
> 3. Thank you for your questions. We believe adopted attacks are appropriate for evaluating OOD safety performance of models:
> i. To ensure a fair comparison with RR and LAT in Table 4, we strictly followed RR's settings, training models from LLaMA-3-8B-instruct, which has undergone extensive safety alignment and training data is likely to cover common OOD attacks [1]. Despite this inherent robustness, our method still achieves measurable improvements, demonstrating its effectiveness.
> ii. Our work focuses on building safe reasoning models from base models, specifically tackling the challenge of generalizing from ID training to OOD attacks—a critical issue highlighted in [2]. Effective attacks often stem from OOD scenarios, as seen in the high ASR of RT models. Moreover, these attacks are widely used in prior studies [3,4] and have proven effective even against aligned models like OpenAI’s ChatGPT.
> 4. Even a few successful attacks are critical, as each vulnerability can be repeatedly exploited by attackers, posing substantial safety risks as discussed in [5].
> For PersuasiveAttack, we strictly followed the original setup. The authors provided only 50 attack questions in their dataset [6].
> For PAIR, we adhered to the original evaluation setup [7], which contains 50 harmful behavior samples. Actually, each sample generates 20 attack prompts, iterating up to 3 times with GPT APIs, producing 1,000–3,000 jailbreak prompts. Evaluating SRG models required ~16 hours, even with fast APIs like GPT-4o-mini (mentioned in Appendix: Lines 712–722).
> 5. We have provided results in 2nd response to Claim part.
> ---
> [1] The Llama 3 Herd of Models.
> [2] Adversarial ML Problems Are Getting Harder to Solve and to Evaluate.
> [3] Refuse whenever you feel unsafe: Improving safety in llms via decoupled refusal training.
> [4] "Do Anything Now": Characterizing and Evaluating In-The-Wild Jailbreak Prompts on Large Language Models.
> [5] https://www.anthropic.com/news/constitutional-classifiers
> [6] https://huggingface.co/datasets/CHATS-Lab/Persuasive-Jailbreaker-Data
> [7] https://github.com/patrickrchao/JailbreakingLLMs

---

> > ### Comment · Reviewer_eX52 · 2025-04-04
> >
> > Thank you for your response! My concerns are addressed. I will raised my score to 3!

---

> > > ### Author Response · Authors · 2025-04-04
> > >
> > > Dear Reviewer,
> > >
> > > Thanks for your reply!
> > >
> > > We appreciate your recognition of our responses. Your comments have been very helpful in refining our revised version, and we will incorporate these discussions accordingly.
> > >
> > > Best regards,
> > >
> > > The authors

---

### Decision · Program_Chairs · 2025-05-01

**Decision:**

Accept (poster)

**Comment:**

This paper examines the limitations of Refusal Training (RT), which often relies on simple pattern matching and struggles to generalize to OOD attacks. To address this limitation, the authors propose a reasoning-based supervision method that guides models to explicitly reason about safety using predefined guidelines.

On the positive side, the paper evaluates SRG across six jailbreak attack types (e.g., JailbreakChat, PAIR) using Attack Success Rate (ASR) as the main metric. Results show that SRG significantly reduces ASR compared to refusal training alone, though it underperforms LAT and RR on certain attacks. The paper also offers interesting insights, as the use of Best-of-N sampling reveals that RT models can generate safe responses when selecting from multiple outputs -- motivating SRG’s approach.

On the negative side, reviewers raised some concerns about novelty---for example, the method combines techniques similar to prior work, particularly CoT distillation used in other domains such as mathematical reasoning. There are also issues with the experiments (choice of baselines, over-reliance on ASR), as well as with the presentation and clarity of the paper.

During the discussion, the authors addressed some of these concerns and provided additional experiments. All reviewers ultimately sided with accepting the paper, and so do I.